# A Beginner’s Guide to the Characterization of Hydrogel Microarchitecture for Cellular Applications

**DOI:** 10.3390/gels8090535

**Published:** 2022-08-26

**Authors:** Francisco Drusso Martinez-Garcia, Tony Fischer, Alexander Hayn, Claudia Tanja Mierke, Janette Kay Burgess, Martin Conrad Harmsen

**Affiliations:** 1Department of Pathology and Medical Biology, University Medical Center Groningen, University of Groningen, Hanzeplein 1 (EA11), 9713 GZ Groningen, The Netherlands; 2W.J. Kolff Research Institute, University Medical Center Groningen, University of Groningen, A. Deusinglaan 1, 9713 AV Groningen, The Netherlands; 3Biological Physics Division, Peter Debye Institute of Soft Matter Physics, Faculty of Physics and Earth Science, Leipzig University, Linnéstraße 5, 04103 Leipzig, Germany; 4Clinic and Polyclinic for Oncology, Gastroenterology, Hepatology, Pneumology, Infectiology Department of Hepatology, University Hospital Leipzig, Liebigstr. 19, 04103 Leipzig, Germany; 5Groningen Research Institute for Asthma and COPD (GRIAC), University Medical Center Groningen, University of Groningen, Hanzeplein 1 (EA11), 9713 AV Groningen, The Netherlands

**Keywords:** extracellular matrix, hydrogel architecture, topography, porosity, electron microscopy, laser microscopy, micro-computed tomography, second harmonic generation, atomic force microscopy

## Abstract

The extracellular matrix (ECM) is a three-dimensional, acellular scaffold of living tissues. Incorporating the ECM into cell culture models is a goal of cell biology studies and requires biocompatible materials that can mimic the ECM. Among such materials are hydrogels: polymeric networks that derive most of their mass from water. With the tuning of their properties, these polymer networks can resemble living tissues. The microarchitectural properties of hydrogels, such as porosity, pore size, fiber length, and surface topology can determine cell plasticity. The adequate characterization of these parameters requires reliable and reproducible methods. However, most methods were historically standardized using other biological specimens, such as 2D cell cultures, biopsies, or even animal models. Therefore, their translation comes with technical limitations when applied to hydrogel-based cell culture systems. In our current work, we have reviewed the most common techniques employed in the characterization of hydrogel microarchitectures. Our review provides a concise description of the underlying principles of each method and summarizes the collective data obtained from cell-free and cell-loaded hydrogels. The advantages and limitations of each technique are discussed, and comparisons are made. The information presented in our current work will be of interest to researchers who employ hydrogels as platforms for cell culture, 3D bioprinting, and other fields within hydrogel-based research.

## 1. Introduction

The ECM is a three-dimensional (3D), acellular, heterogeneous network composed of fibrillar force-transducing collagens, interconnecting proteins such as fibronectin, matricellular proteins (e.g., periostin, fibulins, osteopontin), and the basement membrane proteins collagen type-IV and laminin [1]. Water retention is accomplished primarily by the highly negatively charged glycosaminoglycans (GAGs) or their higher order structures, i.e., GAGs bound to a protein core (proteoglycans), and to a lesser extent by collagens and similar proteins that also retain water [2]. The water concentration is highly tissue-specific, but it can range from 5% to 90% [3]. The ECM provides structural support and instruction to cells governed by its biophysical and biochemical cues.

Among the materials employed to mimic the ECM are hydrogels: highly porous, interconnected, hydrophilic, 3D polymeric networks that absorb and hold over 20% of their mass in water or other biological fluids [4,5]. When loaded with cells, hydrogels can provide biophysical conditions similar to those found in the native ECM [6,7]. For example, cell adhesion is not limited to a single plane, and there is no forced polarity as observed in vivo. Instead, in hydrogels, cell spreading and migration are modulated due to the variable stiffness and viscoelasticity of the material [8,9]. Hydrogels are formed via physical and chemical crosslinks and are commonly classified based on their polymer sources [4,10,11]. Fibrin [12], collagen [13], and decellularized organ-derived ECM [14,15,16,17,18] are examples of natural hydrogels and are regarded as biocompatible and bioactive [19]. These materials retain native cell-binding sites as well as protease-targeted degradation motifs, but due to their sources, batch-to-batch variations can influence the hydrogels’ tunability and overall mechanics and microarchitecture [10,19,20]. Synthetic hydrogels, such as polyacrylamide and polyethylene glycol (PEG), are regarded as more tunable than natural hydrogels, as their reconstitution conditions result in fewer batch-to-batch variations. Modifying the polymer backbones and molecular weights of synthetic hydrogels are common approaches to fine-tuning their properties. However, synthetic hydrogels lack inherent fundamental biological cues and require conjugation with cell-binding peptides (e.g., RGD, GFOGER, and IKVAV) to be biocompatible [10,21,22,23]. Semi-synthetic hydrogels, such as gelatin methacryloyl (GelMA) [24,25,26,27] or methacrylated hyaluronic acid (HAMA) [28,29], stem from the incorporation of crosslinking sites into the backbone of a natural polymer. These sites grant semi-synthetic hydrogels with a mechanical stability and tunability not commonly achieved in natural hydrogels [30,31,32]. Semi-synthetic hydrogels retain some biocompatible and bioactive features, being derivatives of natural polymers. For a more detailed description of the properties of individual hydrogel polymers, the reader is referred elsewhere [4,19,33]. The wide variety of polymers available for cell culture assays allow researchers to tailor hydrogel-based cell culture models to their research question(s). Within hydrogels, the microarchitecture is an inherent property known to influence the cell fate [34,35,36].

### Hydrogel Microarchitecture

In biology, microarchitecture refers to the detailed structure of any organ at a micrometer scale. In hydrogels, the microarchitecture depends on the organization of the polymer network during the sol–gel transition, the polymer concentration, and the crosslinking conditions (e.g., ionic strength, temperature, and pH), among others [37,38]. The resulting polymer network, known as the mesh size or molecular porosity, influences oxygen and nutrient diffusion [39]. Porosity is the percentage of void space in a material and it represents a fraction of the total volume [40,41]. Pore sizes influence contact guidance during cell migration or inhibit cell orientation [42,43]. A lower porosity induces cell aggregation and inhibits proliferation (e.g., 93% vs. 97% porosity) [44]. Large-sized pores may compromise the mechanical stability of the polymer network due to the excessive void, depending on the crosslinks holding the network together [45,46]. In hydrogel-based tissue engineering, pore size is critical for bone (>300 µm) [40,47], (250–500 μm) cartilage [48], and vascular network formation (~166 μm) [49] to occur both in vivo [40,50,51,52] and in vitro [40,47,53]. Nonetheless, such processes are not solely pore size-dependent [45], highlighting the importance of ECM composition [45] and conformation [37] in determining the cell fate. Other components of hydrogel microarchitectures include the fiber diameter, length, and orientation [42,54,55,56], as well as network inhomogeneity [57], which play an active role in cell invasion [58,59,60,61,62]. For example, in a cancer cell model, the cell morphology, cluster formation, and cell invasion were regulated by the fiber diameter (850 nm) and not the pore size (7.5–11 μm) [58].

Cell–matrix interactions can change the hydrogel microarchitecture, with diverse outcomes based on the polymer type [63,64]. Additional microarchitecture parameters, such as surface topography, are mostly described in 2D hydrogel-based cell culture models. Topographic stimuli also influence cell adhesion [65], contact guidance [66,67], migration [65], and overall gene regulation [66,68,69].

Due to its relevance in hydrogel-based research, microarchitecture is of increasing interest to researchers, and the number of publications has substantially grown over the years (Figure 1). The data reported on hydrogel microarchitecture depend on the method employed to assess it, and the systematic errors behind said method need to be carefully considered. Therefore, the present work aims to provide an overview of how those data are produced. Our study’s novel aim is to integrate the known techniques and facilitate the choice of methods by relatively inexperienced investigators in order to improve understandings of ‘cells in gels’ in this multidisciplinary field. Thee accurate rate quantification of the microarchitectural parameters in biopolymer networks is essential for elucidating the observed effects of hydrogels on cell biology. Most methods used for assessing hydrogel microarchitecture can be classified as electron-based or photon-based, based on their underlying principles, and they are further explained in this work.

## 2. Electron-Based Techniques

### 2.1. Scanning Electron Microscopy

Scanning electron microscopy (SEM) is the most widely reported method for characterizing hydrogel microarchitectures [17,20,38,63,70,71,72,73,74,75,76,77]. This high-resolution imaging tool provides a detailed visualization of the hydrogel surface at the nanometer scale [78]. As its name suggests, SEM is an electron-based technique, where a high energy beam (aka “electron gun”) bombards a metal- or carbon-coated specimen with primary electrons, causing the emission of secondary and backscattered electrons. Secondary electrons highlight the morphology and topography of the specimen, while backscattered electrons provide contrast between areas with distinct chemical compositions (Figure 2). SEM imaging occurs under a high vacuum, as the presence of gas can attenuate the electron beams and stop them from scattering [79]. SEM microphotographs of hydrogels are employed to determine pore size, pore distribution, and porosity percentage, as well as fiber thickness and fiber orientation [17,38,63,72,73,74]. In cell-loaded hydrogels, the visualization of the cells is also possible [80,81,82,83,84]. The analytical capabilities of SEM include X-ray-based tools, such as energy dispersive X-Ray spectroscopy (EDX). EDX can detect elements such as C, S, O, N, Na, and others that are present in hydrogels [85,86,87,88] and within cells (e.g., P—a marker of DNA) (Figure 3A). EDX is particularly useful for the recognition of cells present within hydrogels of marked structural heterogeneity (Figure 3B). For more in-depth information on the use of EDX in biomedical research and diagnosis, the reader is referred elsewhere [89].

SEM-generated data indicate that higher polymer concentrations decrease pore sizes, but that cells are capable of modifying such porosity in hydrogels with degradation-sensitive sites [63]. In hydrogels that depend on functionalized groups to form crosslinks, SEM demonstrated that the degree of functionalization (DoF) has a greater influence on the pore density, pore size, and porosity percentage than the polymer concentration [90]. For example, high-DoF hydrogels have smaller pores than low-DoF hydrogels at similar polymer concentrations [30,90]. Moreover, both the polymer concentration and DoF have a direct influence on hydrogel swelling (i.e., water retention) and mechanics [63,74,90,91].

The limitations of SEM arise during the hydrogel preparation steps, as visualization requires a dry specimen. Thus, SEM is inherently biased, as desiccation will alter the native microarchitecture. Hydrogel desiccation is commonly achieved by passing a sample through a gradation of alcohol dehydration series [18,37,75,76,77,92] followed by freeze-drying [38,63,72,74,76,77] or critical point drying [17,37,92]. Thus, desiccation irreversibly alters the microarchitecture, leading to an imprecise hydrogel representation [77]. For example, collagen-HA hydrogels dried at −20 °C, −70 °C, and –196 °C showed variable (mean) pore sizes of 230, 90, and 40 μm, respectively [38]. Methods of applying fixatives such as glutaraldehyde [17,92] or combining with paraformaldehyde have been reported [20,37] for both cell-free and cell-loaded materials, but it is not clear to what extent the artefacts are prevented. Such artefacts can destroy finer features and leach out ions of interest [93]. Hydrogels are non-conductive, requiring irreversible carbon or metal coating (e.g., Au-Pd) [17,20,37,38,74,76] that could conceal finer surface details [93]. Despite these drawbacks, SEM data serve as a comparative measure when all hydrogels sustain the same systematic processing error. Moreover, SEM specimens can be preserved and visualized repeatedly, unlike samples imaged in other electron-based techniques, such as cryogenic SEM (Cryo-SEM) or environmental SEM (ESEM).

### 2.2. Cryogenic Scanning Electron Microscopy

Cryogenic scanning electron microscopy (Cryo-SEM) relies on a standard SEM with a field emission electron gun but employs a cryo-transfer system, where samples can be coated, fractured, and sublimated. In Cryo-SEM, samples must undergo vitrification: an ultra-rapid freezing method that prevents water crystal formation and generates a glass like-specimen [94]. The most widely reported method of vitrification in hydrogels is by plunge freezing either in liquid nitrogen, liquid ethane, liquid propane, or nitrogen slush at −137 °C [77,95,96,97]. Post-vitrification metal coating is not deemed essential but improves the imaging resolution [95]. Specimens can be fractured in order to visualize their inner-most microarchitecture and sublimated to remove the top layer of water, revealing the underlying microarchitecture [98]. The fast freezing step preserves biological structures with a higher fidelity than conventional SEM, rendering the Cryo-SEM imaging more factual in order to evaluate the hydrogel pore size [95], porosity [99], and fiber diameter (Figure 2) [98,99]. The presence of cells can also be detected [100,101]. As in SEM, most limitations of Cryo-SEM arise during the sample preparation stage. While, in principle, vitrification prevents ice crystallization, this process depends on a high cooling rate, which is difficult to achieve in specimens with a >10 µm thickness. The use of high-pressure freezing (Figure 2), which consists of a stream of liquid nitrogen at a rate of >2000 bar (~1.5 × 10^6^ Torr) pressure can reportedly vitrify samples of a ≤500 µm thickness at −196 °C [94]. A poor cryo-fixation generates hexagonal ice crystals that displace the polymer network, causing structural damage [95]. Adding cryoprotectants improves the vitrification process, although the effective concentrations have been reported as cytotoxic upon prolonged exposure [97]. It is unclear whether cytotoxicity would cause any real alterations to cell-loaded specimens, as this step is performed immediately before freezing. Sublimation reportedly caused cracks on the surfaces of alginate hydrogels [95], and these are likely to occur in other polymer networks as well. Slow freezing rates are also reported, while sample dehydration and architecture distortion are common artefacts [95,96]. Compared to SEM, fewer studies report on the use of Cryo-SEM for hydrogels. For a detailed guide on hydrogel preparation for Cryo-SEM, the reader is directed elsewhere [95].

### 2.3. Environmental Scanning Electron Microscopy

ESEM is another electron-based mode, with the particularity that vacuum conditions allow for the presence of water vapor inside the imaging chamber [79]. In ESEM, the electron gun is kept at a short distance from the sample to reduce vapor interference. As secondary electrons are emitted, their collision with gas molecules amplifies the signal detection. Hence, water vapor not only hydrates the sample, but also plays a key role in image generation [79]. Two modalities of ESEM are available, a wet mode and a low-vacuum mode [102]. In the wet mode, the vapor pressure remains between 4–6 Torr, and the sample is kept cool at 5 °C. In the low-vacuum mode, the pressure within the chamber remains at 1 Torr, keeping a 5% relative humidity [103]. These imaging conditions render the specimen desiccation and metal coating unnecessary.

As the specimens remain in their native, swollen state, ESEM is regarded as optimal for biological samples (Figure 2) [79,93,104]. Resolution limits in ESEM are slightly lower than those in SEM (50 nm > 10 nm; respectively), but the preservation of the morphological integrity of the specimen is an advantage [78,79,93]. The use of cross-sections to visualize the internal microarchitecture of hydrogels with ESEM has also been reported [105]. Despite the clear advantages of ESEM compared to the previously mentioned SEM modes, some limitations are involved. Firstly, the high humidity in the chamber can cause water condensation on the sample surface, impeding visualization [78]. Due to the hydrophilic nature of hydrogels, this is likely to occur. Secondly, despite the humid conditions within the chamber, both the constant voltage and changes in vacuum pressure during imaging will invariably induce artefacts [102,106]. Thirdly, imaging must be performed within a relatively short time frame (<45 min), as the voltage and vacuum will inevitably alter the sample. For this reason, ESEM samples require immediate visualization, unlike SEM specimens that can be stored for later imaging. As with Cryo-SEM, ESEM is regarded as underutilized for hydrogel imaging [95,102,106,107], demonstrating a relevant gap in our knowledge of the microarchitecture of polymer networks in their native wet state.

## 3. Photon-Based Techniques

### 3.1. Micro-Computed Tomography

Micro-computed tomography (µ-CT) is an X-ray-based scanning imaging tool that generates 2D trans-axial projections of a specimen [108,109]. While µ-CT is classified in this work as a photon-based technique, X-rays derive from the electron interactions within a high-energy electromagnetic beam [110]. In the µ-CT equipment, the sample is placed on a rotational stage and exposed to an X-ray source, and the passing light is captured by an X-ray detector. The passing X-rays can be attenuated (i.e., absorbed or scattered) by the sample thickness, density, and composition, providing phase contrast to the structures and components [111]. The µ-CT images can be reconstructed in 3D with an up to 1 μm voxel (3D pixel) resolution, making this a high-resolution technique. As an X-ray-based imaging method, µ-CT has historically been reported as a tool for reconstructing bone microarchitecture [112,113,114,115]. Studies on cell-loaded and cell-free hydrogels have used this technique to detect hydrogel mineralization both in vitro and in vivo [116,117,118,119]. The µ-CT data can be used to determine the pore size and fiber thickness in order to reconstruct the polymer network in 3D, demonstrating the pore interconnectivity [102,108]. Hydrogel degradation tests employing µ-CT in vitro demonstrate an increase in the pore sizes and porosity percentages [105,120,121,122]. While µ-CT is regarded as both non-destructive and non-invasive, exposure to high current and voltage levels will invariably dehydrate the sample [123]. High voltage levels are common when generating high-resolution images, but they are costly and result in a prolonged imaging time. To prevent structural damage during imaging, some reports recommend specimen fixation or the use of (freeze-)dried samples [116,121]. Thus, it is not uncommon to validate hydrogel µ-CT data using SEM data derived from dry specimens [105,123,124,125]. There are other limitations of µ-CT, as the hydrophilic nature of hydrogels provides a low-phase separation contrast (Figure 4) [119]. Radiopaque agents can be coupled with the polymer backbone or solubilized and left to soak with the hydrogel to improve said contrast [119]. Contrast agents allow us to discern between the hydrogel fibers and porosity and have been used to investigate hydrogel degradation in vivo and in vitro [126,127]. Reportedly, osmium tetroxide and uranyl acetate, or a combination of uranyl acetate and lead citrate, enabled a good contrast resolution and 3D reconstruction of collagen-I hydrogels [128]. The use of metal nanoparticles (e.g., Au) has also been reported to improve the hydrogel contrast [119] or the visualization of the spatial distribution of the cells within. Other limitations include vibrations in the µ-CT’s rotatory stage, which can cause motion artefacts by displacing the sample if it is not properly fixed—a common challenge with wet materials [123]. Information on the in-depth specifications and further limitations of µ-CT in materials research can be found elsewhere [111,123].

### 3.2. Confocal Laser Scanning Microscopy

Confocal laser scanning microscopy (CLSM) is a photon-based fluorescent imaging technique. For visualization, CLSM requires that proteins and structures are stained or coupled with fluorophores: organic molecules that can emit light (λ_em_) upon light excitation (λ_ex_) [129]. The CLSM optical resolution is determined by the wavelength of the laser. This wavelength is chosen according to the absorption spectra of the fluorophores, and multiple lasers can be used at the same time or in succession to capture multi-color fluorescent images [130]. Unlike conventional fluorescent microscopes that illuminate the entire specimen, CLSM uses point illumination, typically a laser beam, and a pinhole in front of the detector to eliminate most of the background blur, greatly increasing the optical resolution [131]. As only a single point of the specimen is illuminated, the samples must be scanned over a specified region to produce 2D or 3D fluorescent images.

CLSM is an invaluable tool used to visualize the structures of living cells within hydrogels [132]. For the assessment of the hydrogel microarchitecture, these can be fluorescently dyed [58], although certain polymers are naturally auto-fluorescent (e.g., chitosan), allowing dye-free imaging [133]. CLSM yields high-resolution 2D and 3D images (Figure 4), revealing the polymer network microarchitecture in great detail, crucial in the determination of the pore size and porosity of hydrogels in their swollen state (Figure 5) [57,58,59]. Overall, CLSM agrees with SEM in regards to the pore size decreasing with increasing polymer concentrations [58]. CLSM images can be segmented into a polymer phase and a fluid phase [42] (Figure 4, orange). During analysis, spheres can be fitted into the fluid phase (Figure 4, blue), and their diameter is measured to determine the pore diameter. The pore diameter, together with the median values of all the detected pore diameters, is defined as the pore size of a particular sample [42]. Applying the same principle to the polymer phase can determine the average thickness of individual fibers (Figure 5A). Additionally, CLSM images can be used to quantify local deviations in the microarchitecture due to nodes formed during the polymerization of collagen-I hydrogels. Moreover, data from fluorescently labelled collagen-I hydrogels showed that fiber diameter and fiber length are heavily influenced by pH, irrespective of the hydrogel concentration. In contrast, the pore size remained unaffected by pH [58].

Data from CLSM applied to cell-loaded hydrogels can be used to detect local cell-induced microenvironmental changes during migration through an inhomogeneity parameter [57]. This inhomogeneity parameter can explain differences in cell migration that cannot be explained using the pore size and mechanical properties alone. Therefore, elucidating these differences has led to significant insights into the role and adaptation of the microenvironment during cell migration [57].

CLSM has some limitations, the most important being the resolution limit due to optical diffraction limits. CLSM employs lasers with excitation wavelengths of several hundred nanometers, and the excited fluorophores commonly emit light at a higher wavelength. For example, collagen fibers in polymerized matrices possess a wide range of diameters, ranging from below 100 nm [134] to nearly 1 µm [58]. Using a λ_ex_ 561 nm-laser and recording λ_em_ 580 nm, objects as small as 290 nm can be distinguished. Thus, this technique [42] must be considered an overestimation, as the fiber thickness can be below the optical diffraction limit. However, due to multiple post-processing steps, the analysis can be considered as quasi-pixelwise. Another important consideration is the efficacy of the fluorescent dyes used. Poor staining might result in a low signal-to-noise ratio or even the inhomogeneous staining of a hydrogel, resulting in limitations in the dynamic range. Thicker and, thus, brighter fibers may overlay smaller fibrils, or vice versa, with larger fibers losing detail when overexposed. Lastly, the choice of a spherical space estimator might not be suitable for a specific problem. For a robust estimate of the cell migration, spheres can be considered a suitable shape. However, other problems can make it necessary to choose other shapes, such as cubes. Fitting shapes into the hydrogel segmentation might be entirely superfluous, and polymer content calculations based on the precise segmentation may provide a better parameter.

### 3.3. Second Harmonic Generation

The higher-order assembly of structural elements (fibrils and fibers) within hydrogels can be examined using non-destructive laser-scanning optical multiphoton microscopy methods [135]. Such techniques are non-linear optical methods that employ femtosecond pulses from near-infrared (NIR) lasers, which enables researchers to examine samples with deeper tissue penetration in comparison to CLSM [136]. The interaction of the femtosecond pulsed NIR light with collagen fibers produces second harmonic generation (SHG) and two-photon excited fluorescence (TPF) signals [137]. The non-absorptive photon interaction with the collagen fibers produces photons with exactly twice the original energy, resulting in an SHG signal that is emitted at exactly half the excitation wavelength. The output signal is dependent on the non-centrosymmetric assembly of the collagen triple helices (secondary structure) and also on the molecular packaging and arrangement of the fibrils (tertiary structure) and the organization of the fibers (quaternary structure) [138]. The TPF signal, on the other hand, is generated through an absorptive process that results from the excitation of inherent fluorophores within the collagen structure (intramolecular pyridinium-type crosslinks and other fluorophores of an unknown nature) [139]. These SHG signals are generated through the specific structural organization of collagens and, therefore, they do not capture information about other mechanically and structurally important elements within the matrix, such as elastin, proteoglycans, and other non-collagenous components. The significant advantage of SHG microscopy is that the signals are generated by the macromolecules, thus obviating the need for staining with fluorophores. SHG has been used to characterize the fiber structural arrangement in collagen hydrogels up to a depth of 1 mm [135,138], and the fiber arrangement, orientation, thickness, and anisotropy have been assessed within hydrogels [140]. Different structural features in collagen hydrogels of a constant concentration that were polymerized under varying pH or temperature conditions can be detected using SHG imaging [138,141]. The measured fiber diameters from SHG images correlate linearly with those from SEM images of the same preparations, although the SEM-calculated diameters are usually smaller due to the dehydration that takes place during the sample preparation [135]. The signal-to-noise ratio for SHG is higher than that observed with TPF, likely due to the quadratic signal of SHG versus the direct concentration dependence of the TPF signal, coupled with the weak auto-fluorescent signal from immature, weakly cross-linked collagen and potentially other ECM molecules, when they are present. This means that homogeneous and incoherent signal emissions can be observed in all fiber orientations using TPF; however, the coherent nature of the SHG signal prevents emission detection when the collagen fibers are orientated parallel to the laser direction [142]. The SHG and TPF signals can provide different information about the structural properties of the collagen hydrogels, e.g., heavily cross-linked collagen in fibrotic tissue and newly deposited collagen. When collagen hydrogels are polymerized under a decreasing temperature, a relationship between the SHG signal and the mechanical properties of the hydrogel can be observed. As the polymerization temperature decreases, the collagen fiber diameter and the pore size detected increase, while the storage modulus G’ and the mean SHG signal decrease. When the crosslinking within the hydrogel is altered using glutaraldehyde, the SHG signal does not change; however, the TPF signal and the storage modulus increase in line with the degree of crosslinking [138]. These data indicate that the SHG and TPF signals impart different information about the characteristics of the collagen hydrogels that correlates with the hydrogel mechanical properties. The detection of forward versus backward SHG signals can provide additional information related to the organization of collagen fibers [143]. Collagen fibers of approximately the same size as the SHG wavelength generate a signal exclusively in the forward direction [136,137,140,144]. In contrast, sites where the fiber thickness changes or fiber interfaces can change the direction of the emitted SHG signal and generate a backward signal [145]. In cell-free collagen hydrogels, any backward SHG signal detected is the result of the scattering of the forward SHG signal, with a small component of the backwards-generated SHG being generated from small fibrils (diameters ~10% of λ2ω). In collagen hydrogels generated at 27 °C, the degree of the backscattering of the SHG signal to the forward detector increases as the initial hydrogel concentration increases, while for hydrogels assembled at 37 °C, the degree of backscattering is much less, reflecting the shorter, more uniform collagen fiber assembly [135]. SHG signals can also be used to investigate how cells remodel their immediate microenvironment when seeded in collagen hydrogels. Cells seeded in collagen hydrogels remodel and contract the hydrogel, altering the collagen microarchitecture [64,146,147]. The SHG signals indicate that the distinct collagen microstructural properties are still present within the hydrogel even after the gel has contracted, with the signal intensity increasing linearly, reflecting the increased collagen concentration as the hydrogels contract [147]. Overtime, cells can remodel the collagen environment within the hydrogels. The SHG signal indicates that the pore size reduces as the collagen bundles (and presumably other ECM components) become larger. SHG has also been used to investigate collagen remodeling and the role of enzymes released from the cells in crosslinking collagen fibers [148]. Collagen fibers that are newly deposited by cells encapsulated in alginate hydrogels are also readily visualized by SHG [149].

### 3.4. Atomic Force Microscopy

Atomic force microscopy (AFM) began as a device capable of measuring forces as small as attonewtons [150]. This technique was advanced to record surface heightmaps on the atomic level, resulting in precise measurements of the surface topology of soft samples [151]. This method, also called scanning force microscopy (SFM) [152], eventually became an invaluable tool in biophysics [153]. AFM can demonstrate the hydrogel surface topology, but also the mechanical properties, through the mathematical modelling of the force–distance curves [148,152]. Hence, AFM can determine the cell elasticity [62] and elastic/Young’s moduli of hydrogels [57]. The AFM working principle is illustrated in Figure 6A-B, and a topological heightmap of a collagen-I hydrogel is depicted in Figure 6C. AFM sensitivity can include a phase separation analysis, which distinguishes between the distributions of different polymers in co-polymer hydrogel formulations (i.e., more than a single polymer type in a single hydrogel) [108].

AFM can also determine the elastic properties of soft matter, such as hydrogels [58] (Figure 6D), by fitting the cantilever-derived curves (Figure 6E) in different models (e.g., the Hertz model) to calculate the elastic modulus of the hydrogels [57,62], such as collagen type-I (Figure 6F). Hence, the AFM serves not only to characterize the surface topology of a hydrogel, but also to determine its elastic/Young’s modulus (i.e., stiffness). However, using the AFM technique to determine the elastic moduli of hydrogels also has limitations. Depending on the actual stiffness of the material, a cantilever with a certain spring constant must be chosen. For example, using a stiff cantilever to probe a soft material will result in a poor signal-to-noise ratio, while using a soft probe for a soft material leads to an optimal signal-to-noise ratio. Vice versa, a soft cantilever might not indent a stiff material.

Depending on the type of hydrogel, the AFM technique can be difficult or even impossible to carry out. Spheroidal probes might stick to the material, resulting in artefacts, disrupted force–distance curves, or even damaged cantilevers. If the cantilever probe is much smaller than the pore size of the hydrogel, it might get stuck in the fibrous network microarchitecture. However, a larger and heavier spheroid influences the spring characteristics of the cantilever. Lastly, the AFM device, in general, is prone to errors due to vibrations, resulting in signal noise and ultimately uncertainties in the Hertz model fit. The usage of vibration dampeners and casings drastically reduces vibrations and are highly recommended. These limitations must be carefully considered when choosing the correct settings to obtain precise measurements.

## 4. Hydrogel Microarchitecture and Cells: Design and Applications

Hydrogels are 2D and 3D cell culture platforms that mimic the ECM and are used in 3D bioprinting and the field of tissue engineering and regenerative medicine (TERM) for cell and drug delivery [8,11,85,154,155]. These cellular applications demonstrate the versatility of hydrogels, and therefore, great efforts are being put into moderating their properties, including their microarchitecture. The hydrogel microarchitecture is described in terms of porosity (percentage), pores (distribution, geometry, interconnectivity, and size), and fiber (thickness/diameter, directionality, and length), as well as surface topography. Researchers have recognized the relevance of these microarchitectural components in driving the cell fate and have aimed to modulate them [53,156,157].

The hydrogel microarchitecture can be modulated by manipulating the crosslinking conditions [158] and by adding porogens, such as salt leaching (e.g., NaCl, NaOH) [159] and gas foaming [160]. Other approaches include 3D printing [161], micropatterning [162], and micro-molding [163] to control the hydrogel geometry and desired topography. Ironically, research citing methods for controlling the hydrogel porosity has assessed this property mostly with SEM [156,157], a biased technique, as previously discussed.

Vascularization and cartilage and bone formation are processes of interest in TERM, which are closely linked to hydrogel microarchitecture. Endothelial cell migration and tissue vascularization were considered optimal in HA-methacrylated (HA-MA) with pore sizes of 200–250 μm [164]. PEG hydrogels with pore size ranges of 50–150 μm facilitated vascularization [165], while GelMA with pores of 49.7 ± 11.8 μm showed capillaries in vitro [62].

PGA scaffolds with a 97% porosity and fiber thickness of 13–15 μm exhibited chondrocyte differentiation [53]. Meanwhile, in genipin-crosslinked gelatin scaffolds, pore sizes between 250–500 μm promoted ECM deposition and chondrocyte proliferation [48].

The osteoblastic differentiation of pluripotent stem cells was reported in hyaluronic acid (HA) hydrogels loaded with bone morphogenetic protein-2 (BMP-2) with pore sizes ranging from 100–600 μm [166]. Human bone marrow stem cells underwent osteoblastic differentiation in silk fibroin scaffolds with ~92% porosity and pore sizes of 920 ± 50 μm [167]. Contrastingly, polycaprolactone (PCL) scaffolds with pore sizes between 300–900 μm had limited effects on the promotion of bone regeneration in vivo [45].

In cancer cell models, cell morphology, cluster formation, and cell invasion are regulated by the fiber diameter (850 nm) and not the pore size (7.5–11 μm) [58]. Thus, the polymer chemistry and the cellular applications, whether in vivo or in vitro, can yield contrasting outcomes. Regardless, the microarchitectural parameters reported here were characterized by the electron-based or photon-based imaging techniques, as previously described. The data generated with said techniques can result in diverse post-processing analyses, some of which can be consulted here [168,169,170].

## 5. Conclusions

Mimicking the ECM through the use of hydrogels is an experimental model of increasing interest. In this review, we have highlighted the microarchitecture data that can be acquired by distinct imaging methods. It is important to mention the following: Firstly, most of the characterization thus far has been performed on cell-free hydrogels. This means that we lack an overview of the cell-induced microarchitectural modifications of hydrogels over time. Secondly, the hydrophilic nature of hydrogels poses a real challenge for the characterization of their microarchitecture, topography, and other mechanical parameters briefly mentioned here, such as stiffness (i.e., elastic/Young’s modulus) and viscoelasticity [91,95]. Thirdly, many of the described techniques are considered non-destructive, but on closer inspection, this only applies to the test phase and not to the sample preparation, which can often lead to significant changes in the hydrogels. Finally, the limitations reported here are not solely technical, but also pertain to the inconsistent interpretation and reporting of the applied parameters [112,171]. There is a clear need to standardize the minimum reporting criteria employed for data acquisition. Thus far, the absence of clear concept definitions, alongside poorly described methods, hinders experimental reproducibility. Adequate reporting, as well as the development of novel biophysical tools, will lead to a deeper understanding of cell–matrix biology in hydrogel systems.

## Figures and Tables

**Figure 1 gels-08-00535-f001:**
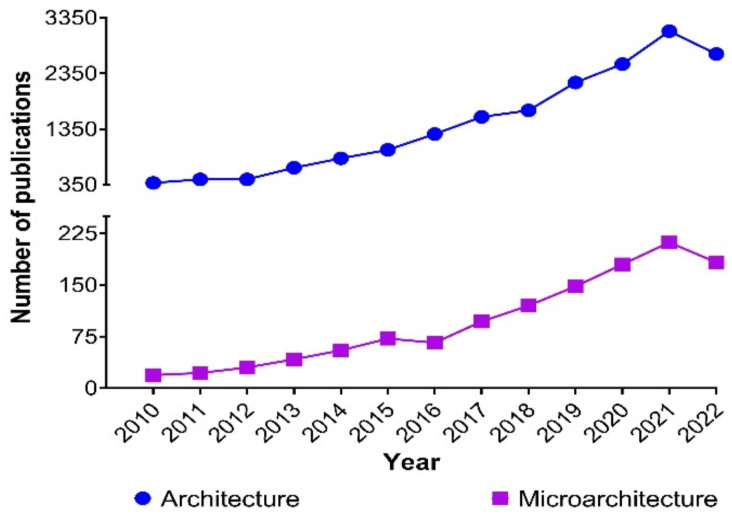
Publications per year from 2010 to June 2022 in the SCOPUS^®^ database (accessed on 21 June 2022), including review articles, research articles, book chapters, conference abstracts, book reviews, conference papers, editorials, mini-reviews, and short communications in all areas of knowledge, using the keywords “hydrogel architecture” (*n* = 18,751) and “hydrogel microarchitecture” (*n* = 1246).

**Figure 2 gels-08-00535-f002:**
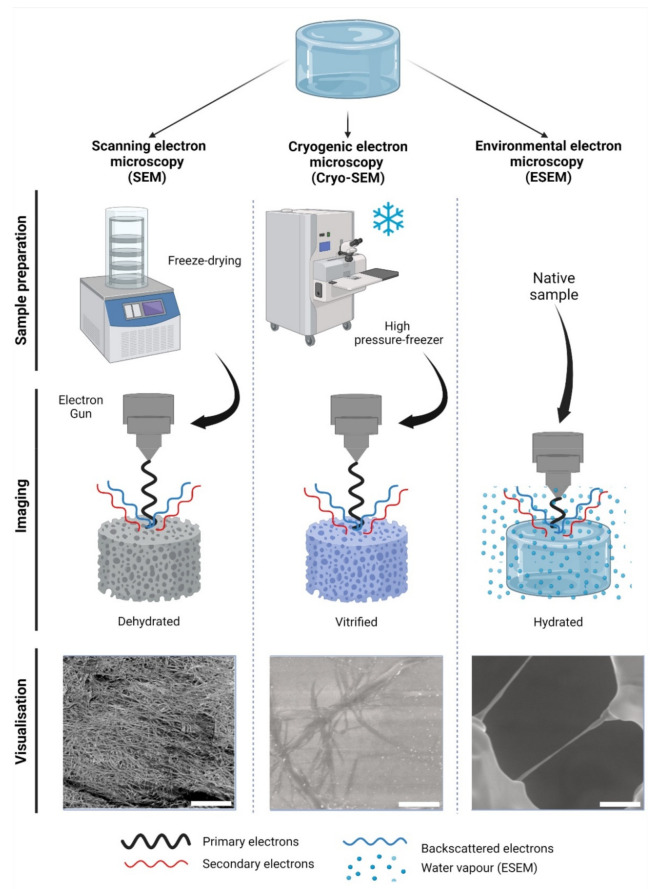
Electron-based imaging techniques for hydrogels. Standard scanning electron microscopy (SEM) relies on sample dehydration, including freeze-drying (shown) after fixation with aldehydes, followed by metal coating. Due to dehydration, the network appears condensed in visualization. In cryogenic SEM (Cryo-SEM), samples are vitrified using—among other methods—a high-pressure freezer (shown). Solid water ice is a source of imaging errors and is seen during imaging. In contrast, environmental SEM (ESEM) does not require a particular sample preparation, as it remains hydrated within a humidified chamber. In ESEM, the electron gun is closer to the specimen than in SEM or Cryo-SEM during imaging. As shown, single collagen fibers can be visualized; however, water condensation can cause imaging artefacts. The visualization of collagen type-I hydrogels (3.0 g/L) is shown at 12,000× magnification, 5 kV, and z = 9 mm. Scale bars represent 5 µm. Collagen type I hydrogels preparation, SEM, Cryo-SEM and ESEM detailed in Appendix A.

**Figure 3 gels-08-00535-f003:**
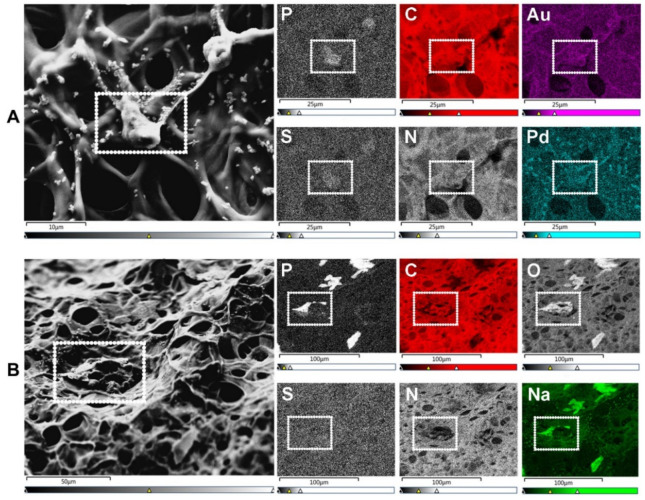
SEM-energy dispersive X-ray spectroscopy (EDX) elemental mapping. Images show human umbilical vein endothelial cells (HUVECs) on the surface of a gelatin methacryloyl (GelMA) hydrogel. (**A**) SEM allows for the visualization of cells (surrounded by a white dotted rectangle), while EDX identifies the elements present in both cells and hydrogels. These elements include non-metals such as phosphorus (P—a marker of DNA/nuclei), carbon (C), sulfur (S), and nitrogen (N). Transition metals used for coatings, such as gold (Au) and palladium (Pd), are also identified with EDX. Scale bars represent 10 µm and 25 µm. (**B**) In specimens with marked heterogeneity, EDX facilitates the identification of cells on the hydrogel surface (white dotted rectangle) that otherwise would not be distinguishable due to the condensation of the polymer network. Additional elements relevant to cell biology include non-metals, e.g., oxygen (O), and alkali metals, e.g., sodium (Na). Scale bars represent 50 µm and 100 µm. HUVEC’s culture conditions and GelMA properties detailed in Appendix A.

**Figure 4 gels-08-00535-f004:**
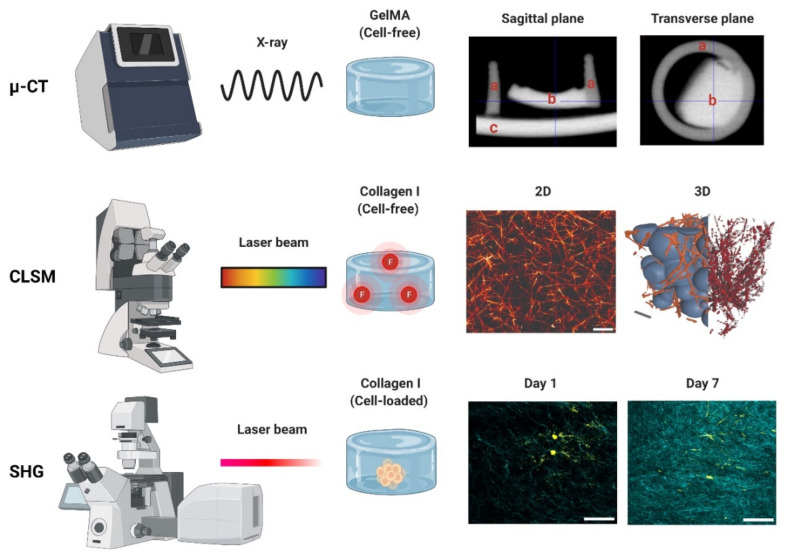
Photon-based imaging techniques. Micro-CT (µ-CT) employs X-rays, creating contrast based on the material properties (e.g., thickness, composition). A GelMA hydrogel scanned with µ-CT is shown in both the sagittal and transverse planes: (a) indicates the PCL container used for casting (2 mm diameter), (b) the GelMA hydrogel, and (c) the stage holder. Confocal laser scanning microscopy (CLSM) employs a laser to excite a fluorophore, which emits the fluorescent signal used for detection. Images of a collagen type-I hydrogel (3 g/L) are shown in 2D and 3D. The 3D image can be used to determine the interconnectivity of the polymer network (orange/red) using a bubble analysis (blue). Scale bars represent 5 µm (2D) and 10 µm (3D). Second harmonic generation (SHG). An example of a cell-loaded (fibroblast) collagen type-I hydrogel is shown on day 1 and day 7. The increase in the SHG signal indicates an increase in the collagen deposition during cell culture. SHG data derived from paraffin-embedded formalin-fixed hydrogels. Scale bars represent 50 µm. µ-CT, CLSM and SHG used to generate the data detailed in Appendix A.

**Figure 5 gels-08-00535-f005:**
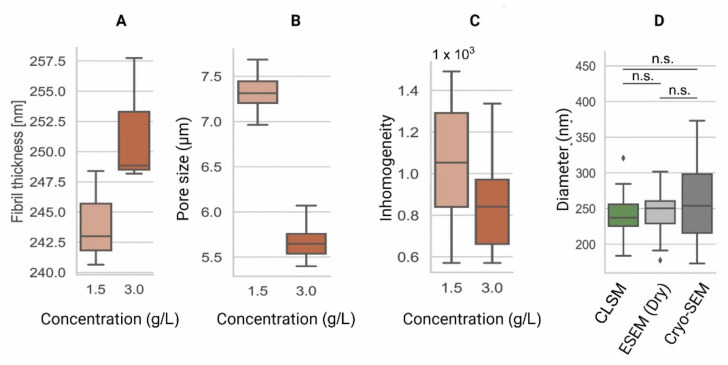
Comparison of the microarchitecture parameters among collagen type-I hydrogels of 1.5 g/L and 3.0 g/L. (**A**) Fiber thickness is an estimation of the 3D cross-sectional diameter of collagen fibers [62]. (**B**) Pore size represents the contiguous 3D space between collagen fibers [42]. (**C**) Inhomogeneity derived from the CLSM data is a measure of local and global variance in the pore size [57]. (**D**) Comparison of pore diameters among CLSM [42], ESEM, and Cryo-SEM images of collagen type-I hydrogels (3.0 g/L). These data indicate similar fiber diameter results among all techniques. Data shown in box-and-whisker plots indicating the median, first and third quartiles as boxes, variability as whiskers and outliers shown as ◆. n.s. = no significant differences.

**Figure 6 gels-08-00535-f006:**
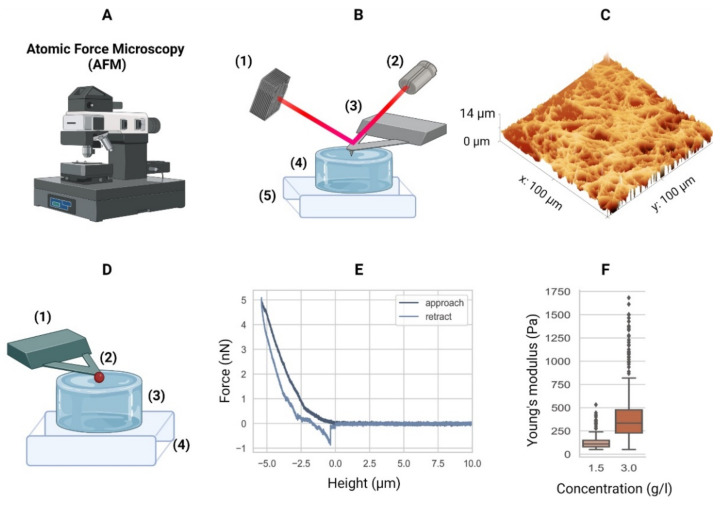
Atomic force microscopy. (**A**) Equipment. (**B**) Schematic of an AFM setup with a four-quadrant photodiode (1), in which the four-quadrant photodiode (1) receives a laser (2) reflected from a cantilever (3), in this case positioned over a hydrogel (4) mounted in a piezo stage (5). For example, the height differences in a sample (4) are measured by adjusting the stage using piezo elements (5) to counter the cantilever bending on a nanometer scale. (**C**) The AFM can then generate a surface heightmap of the hydrogels such as a GelMA hydrogel (shown). AFM can also be used to determine the mechanical properties of hydrogels. (**D**) Schematic of the AFM technique to determine the elastic moduli of hydrogels with a tipless cantilever (1), spheroidal probe (2, red), hydrogel (3), and stiff substrate (4). As the cantilever represents a spring with a known spring constant, the cantilever bending due to elastic counterforces exerted by the soft material is correlated with the piezo stage height (4). (**E**) The so-called force–distance curves are recorded. Data from a collagen type-I hydrogel (3.0 g/L) are shown. (**F**) Young’s moduli of a 1.5 g/L and 3.0 g/L collagen type-I hydrogel. Outliers indicated by ◆. AFM equipment detailed in Appendix A.

## Data Availability

All data presented in this work are available upon reasonable request.

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
