# Peer review of "A Beginner’s Guide to the Characterization of Hydrogel Microarchitecture for Cellular Applications"

_gels, 2022, doi:10.3390/gels8090535_

Round 1

Reviewer 1 Report

The article lacks a discussion on the different types of hydrogels made with different polymers, and manufacturing techniques. Although the authors explain the fundamentals of these techniques, it lacks discussion with the most current references.

1.      Reviewer comments: The number of new publications in the field is high and growing day by day. But it is also true that the number of good reviews in the area is equally increasing. Hence, I believe new reviews should be focused on the recent advances while making use of the efforts from previous reviews to substantiate the knowledge in the field. The authors can add a statistical graph of the articles published in the last 10 years, referring to hydrogels and their application for cellular applications.

2. Reviewer comments: It is still difficult to find the novelty of the work concerning what has already been published. A literature review is required. What is the difference between what is published with what the authors want to publish? It is not clear. The authors must describe these differences and must be described in the introduction section.

3.   Reviewer comments: What do the authors refer "cellular applications"?. Explain more in detail.

4.   Reviewer comments: The authors must add a section dedicated to cellular applications. and describe the current challenges. Also, the authors must explain the physicochemical, mechanical, and biological properties that must have the hydrogels for cellular applications. Explain more in respect in the section.

5.    Reviewer comments: Line 37. “The ECM is a three-dimensional (3D) acellular, heterogeneous network composed mainly of fibrous proteins and proteoglycans, saturated with water”. What type of proteins? What water concentration????

6.  Reviewer comments: Line 40. The authors must improve the definition of hydrogels. The authors can use the cite: 10.1016/j.eurpolymj.2020.110176.

7.      Reviewer comments: Line 75. “Porosity is the percentage of void space in a material and it represents a fraction of the total volume”. What average values are the most appropriate for cellular applications.? Describe examples of hydrogels with these values and their main conclusions using cells.

8.      Reviewer comments: Line 75. “Pore sizes influence contact guidance during cell migration or inhibit cell orientation. Small size pores induce cell aggregation and inhibit proliferation. Large size pores may compromise the mechanical stability of the polymer network due to excessive void, depending on the crosslinks holding the network together”. The authores must describe that values can be considered as small and large size pores and describe examples of hydrogels with these size pores for cellular applications.

9.      Reviewer comments. Line 80. In hydrogel-based tissue engineering, pore size is critical for bone and cartilage and vascular network formation to occur both in vivo and in vitro. What type of cells is present in bone, cartilage, and vascular tissue? What pore size is required?

10.  Reviewer comments. Figure 1 must contain examples of SEM image where porosity and pore size is best represented. Also, where are the references?

11.  Reviewer comments. Where are the references in all the Figures? Permission???

12.  Reviewer comments. The authors must explain the importance of each characterization method described in the manuscript. Although the methods evaluate the structural properties, each technique has a particular purpose in hydrogels.

Author Response

We thank reviewer 1 for time and efforts invested in commenting our manuscript. We have addressed each point raised below:

The article lacks a discussion on the different types of hydrogels made with different polymers, and manufacturing techniques. Although the authors explain the fundamentals of these techniques, it lacks discussion with the most current references.

Thank you for the observation. It was not our goal to discuss manufacturing techniques of distinct hydrogels but to review analyses methods. Manufacture of hydrogels would be topic of a different review.

  1. Reviewer comments:The number of new publications in the field is high and growing day by day. But it is also true that the number of good reviews in the area is equally increasing. Hence, I believe new reviews should be focused on the recent advances while making use of the efforts from previous reviews to substantiate the knowledge in the field. The authors can add a statistical graph of the articles published in the last 10 years, referring to hydrogels and their application for cellular applications.

We agree with the reviewer. The suggestion to include statistical data is relevant to substantiate our work. Hence, we included a similar chart in the introduction, using the reference provided by the reviewer (10.1016/j.eurpolymj.2020.110176) as a guide.

  1. Reviewer comments:It is still difficult to find the novelty of the work concerning what has already been published. A literature review is required. What is the difference between what is published with what the authors want to publish? It is not clear. The authors must describe these differences and must be described in the introduction section.

Thank you for the observation. Many reviews on hydrogels largely describe the materials and their biological and chemical properties. Hydrogel microarchitecture is known to drive cell fate but no reviews have actually touched upon how those data are produced and which techniques are employed. Moreover, most describe said properties overlooking the influence of cells, which we have touched upon – although the data is scarce when it comes to correlating cellular modification to hydrogel microarchitecture. Our novelty is to integrate the known techniques and facilitate choosing methods to understand ‘cells in gels’ by relatively inexperienced investigators in this multidisciplinary field. We have clarified this within the Introduction.

  1. Reviewer comments:What do the authors refer "cellular applications"?. Explain more in detail

We have expanded on this concept in a new section of our manuscript (4.1. Hydrogel microarchitecture and cells: design and applications).

  1. Reviewer comments: The authors must add a section dedicated to cellular applications. and describe the current challenges. Also, the authors must explain the physicochemical, mechanical, and biological properties that must have the hydrogels for cellular applications. Explain more in respect in the section.

As suggested by the reviewer, we have expanded on the concept of cellular applications by creating a new section (4.1. Hydrogel microarchitecture and cells: design and applications).

  1. Reviewer comments:Line 37. “The ECM is a three-dimensional (3D) acellular, heterogeneous network composed mainly of fibrous proteins and proteoglycans, saturated with water”. What type of proteins? What water concentration????

Thank you for the observation, we have clarified we refer to fibrillar force-transducing collagens, interconnecting proteins such as fibronectin, matricellular proteins and the basement membrane proteins collagen type IV and laminin. Water retention is accomplished primarily by the highly negatively charged glycosaminoglycans (GAGs) or their higher order structures i.e. GAGs bound to a protein core (proteoglycans) and to a lesser extent collagens and similar proteins also retain water. Water concentration varies per tissue, ranging from 5-80%.  

  1. Reviewer comments: Line 40. The authors must improve the definition of hydrogels. The authors can use the cite: 10.1016/j.eurpolymj.2020.110176.

We thank the reviewer for the suggestion, we have expanded our definition based on the references provided.

  1. Reviewer comments: Line 75. “Porosity is the percentage of void space in a material and it represents a fraction of the total volume”. What average values are the most appropriate for cellular applications.? Describe examples of hydrogels with these values and their main conclusions using cells.

We have expanded as suggested on section 4.1. We have also included said values when reported.

  1. Reviewer comments: Line 75. “Pore sizes influence contact guidance during cell migration or inhibit cell orientation. Small size pores induce cell aggregation and inhibit proliferation. Large size pores may compromise the mechanical stability of the polymer network due to excessive void, depending on the crosslinks holding the network together”. The authors must describe that values can be considered as small and large size pores and describe examples of hydrogels with these size pores for cellular applications

We thank the reviewer for pointing this out. We have included the values when reported in the references provided.

  1. Reviewer comments. Line 80. In hydrogel-based tissue engineering, pore size is critical for bone and cartilage and vascular network formation to occur both in vivo and in vitro. What type of cells is present in bone, cartilage, and vascular tissue? What pore size is required?

We have expanded on these biological events on the before mentioned section, 4.1.

  1. Reviewer comments. Figure 1 must contain examples of SEM image where porosity and pore size is best represented. Also, where are the references

Under SEM, the collagen hydrogel gets condensed due to the dehydration and freeze drying steps (irrespective of fixation with e.g. glutaraldehyde), and thus, porosity and pore sizes are not visible or at least difficult to discern. The goal of figure 1 is to show side by side, a representative microphotograph of each EM technique from near identical samples, at the same magnifications. Hence, it would be biased to intend to demonstrate a property that is not visible with this technique.

  1. Reviewer comments. Where are the references in all the Figures? Permission??

All data are original i.e. generated by our own groups and not published elsewhere.

  1. Reviewer comments. The authors must explain the importance of each characterization method described in the manuscript. Although the methods evaluate the structural properties, each technique has a particular purpose in hydrogels.

Thank you for the suggestion. We have expanded on additional uses of these techniques when applied to hydrogels. In particular, CLSM is also employed to observe cells in gels and this has been clarified.

Reviewer 2 Report

In the review paper "A beginner’s guide to characterizing the structural properties of hydrogels for cellular applications", the authors describe and compare the most widely used techniques for characterization of the microstructure of hydrogel materials. The title of the manuscript, however, is misleading, as it claims being a guide for characterizing structural properties of hydrogels but solely focuses on the visualization of samples using microscopy and tomography techniques. Structural properties would also include hydrogel mechanics, swelling, crosslinking kinetics and density, and polymer conformation (crystallinity), requiring the use of many other techniques including mechanical testing, rheology, FTIR spectroscopy etc.  Therefore, the title and the abstract of the paper should be revised accordingly. 

Other comments:

* The sections of the review paper are based on the listing of individual imaging methods and their description. It would be more practical and easier to follow if the sections were based on the microstructure of the hydrogels such as porosity, fiber dimension, mineralization, elemental composition etc., describing their importance on cellular applications and listing specific analysis techniques and methods for each property.  

* Porosity/pore size and fiber dimensions are mentioned many times in the text, but the methods of quantification such as the use of digital analysis software ImageJ have not been described. It would be more helpful for the reader if quantification methods are briefly described.

* SEM methods mainly focuses on surface imaging of the hydrogels, while the bulk properties are more important for cell encapsulation applications. More insight into the characterization of hydrogel cross-sections and methods such as cryo-sectioning should be discussed. 

* The manuscript  describes only the characterization of macroscale gels. Hydrogels are also utilized in nano- and microscale for the encapsulation and delivery of cells or drugs by fabrication through microfluidics, electrospraying, emulsification etc. The manuscript should also mention the characterization of nano- and microgels for morphology (microscopy), mechanics (AFM) etc. 

* Figure legends lack the source of data presented. Are they original images from the researchers or retrieved from other studies?

Author Response

We thank reviewer 2 for the time and efforts invested in our manuscript. We have addressed each point raised below:

In the review paper "A beginner’s guide to characterizing the structural properties of hydrogels for cellular applications", the authors describe and compare the most widely used techniques for characterization of the microstructure of hydrogel materials. The title of the manuscript, however, is misleading, as it claims being a guide for characterizing structural properties of hydrogels but solely focuses on the visualization of samples using microscopy and tomography techniques. Structural properties would also include hydrogel mechanics, swelling, crosslinking kinetics and density, and polymer conformation (crystallinity), requiring the use of many other techniques including mechanical testing, rheology, FTIR spectroscopy etc.  Therefore, the title and the abstract of the paper should be revised accordingly. 

We thank the reviewer for the observation. We agree that the word ‘structural’ hints to other hydrogel properties such as elastic and viscoelastic properties as well as gelation kinetics. Indeed, all these properties are intertwined within hydrogel structure. As we provide a definition of microarchitecture and what it means for hydrogels, we have rephrased the title of our manuscript accordingly. Other techniques mentioned by the reviewer do not serve to assess the hydrogel microarchitecture and therefore, have been excluded from our review.

Actually, the reviewer touches on a relevant issue which is the lack of proper terminology for hydrogels or, for that matter, confusion between different research areas on terms like ‘architecture’ and ‘structure’ – for a biologist, chemist or physicist these would all have different meanings. In due time, some form of standardization is warranted.

Other comments:

* The sections of the review paper are based on the listing of individual imaging methods and their description. It would be more practical and easier to follow if the sections were based on the microstructure of the hydrogels such as porosity, fiber dimension, mineralization, elemental composition etc., describing their importance on cellular applications and listing specific analysis techniques and methods for each property.  

Our initial drafts were composed as the reader suggests but generated more confusion than clarity. Therefore, we kindly disagree with the reviewer’s request. Moreover, the goal of the review was to focus on said techniques, not the properties themselves. The main reason being is said properties will change even within the same sample by varying single steps of the sample prep method (e.g. found in Line 146: For example, collagen-HA hydrogels dried at −20°C, −70°C and –196°C showed variable (mean) pore size of 230, 90 and 40 μm, respectively [35]). Such systematic errors arise due to the technique employed. Therefore, our naim was to emphasize those limitations. To date, no review covered such information on hydrogels, let alone, referencing cell-loaded materials. We have expanded the cellular applications concept by reporting microarchitectural parameters.

* Porosity/pore size and fiber dimensions are mentioned many times in the text, but the methods of quantification such as the use of digital analysis software ImageJ have not been described. It would be more helpful for the reader if quantification methods are briefly described.

We have added a section on the cellular relevance of these properties and have touched upon the existing methods to quantify them in our new section: 4.1. Hydrogel microarchitecture and cells: design and applications

* SEM methods mainly focuses on surface imaging of the hydrogels, while the bulk properties are more important for cell encapsulation applications. More insight into the characterization of hydrogel cross-sections and methods such as cryo-sectioning should be discussed. 

We kindly disagree with the reviewer: obviously by definition SEM generates images from (metal)coated materials. Yet, depending on the preparation of the specimens this could either be ‘looking’ inside gels (after cutting in halfs) or indeed ‘looking’ only at the surface. As it appeared after our literature search and talking to other specialists in the field of cryo-sectioning (we think the reviewer meant cryo-SEM?) and based on our own experience, cryo-based methods fall short of reproducibility, accuracy and resolution. While compared to most other discussed techniques the cryo-methods require the highest skills level. Therefore, we did not add it to the review.

* The manuscript describes only the characterization of macroscale gels. Hydrogels are also utilized in nano- and microscale for the encapsulation and delivery of cells or drugs by fabrication through microfluidics, electrospraying, emulsification etc. The manuscript should also mention the characterization of nano- and microgels for morphology (microscopy), mechanics (AFM) etc. 

Indeed, hydrogels have applications beyond those of the main area of interest of this review, but characterisation of the microarchitecture of nano and microscale still relies on the techniques already described.

* Figure legends lack the source of data presented. Are they original images from the researchers or retrieved from other studies?

Thank you for the observation. All data presented here is original and unpublished elsewhere.

Round 2

Reviewer 1 Report

The authors adequately answered each of the questions. The article can be accepted once minor changes have been taken into account.

1. The authors must describe in more detail each of the figures.

Author Response

We would like to thank the reviewer again for the time invested in our manuscript. We have expanded the description of the figures both in the captions and in a new section, namely Appendix A. We believe this information will bring clarity to the data presented.

Francisco Drusso Martinez Garcia

Reviewer 2 Report

I would like to thank the authors for kindly addressing the comments in the revised manuscript.

Author Response

On behalf of all the authors, I would like to thank the reviewer again for their time invested in our manuscript and for helping us improve the quality of our work.

Kind regards

Francisco Drusso Martinez Garcia